# Antioxidant Potential of Flower Extracts from *Centaurea* spp. Depends on Their Content of Phenolics, Flavonoids and Free Amino Acids

**DOI:** 10.3390/molecules26247465

**Published:** 2021-12-09

**Authors:** Piotr Salachna, Anna Pietrak, Łukasz Łopusiewicz

**Affiliations:** 1Department of Horticulture, West Pomeranian University of Technology in Szczecin, 71-459 Szczecin, Poland; anna.pietrak@zut.edu.pl; 2Center of Bioimmobilisation and Innovative Packaging Materials, West Pomeranian University of Technology in Szczecin, 71-270 Szczecin, Poland; lukasz.lopusiewicz@zut.edu.pl

**Keywords:** *Centaurea*, antioxidants, plant natural products, antioxidant properties

## Abstract

Scientists intensely search for new sources of antioxidants, perceived as important health-promoting agents. Some species of the large genus *Centaurea* provide raw materials for the pharmaceutical and cosmetic industries, as well as produce edible flowers. This is the first study that determines the content of total polyphenols, flavonoids, reducing sugars, free amino acids and the antioxidant potential in the flower extracts of *C. nigra* L., *C. orientalis* L. and *C. phrygia* L. The total polyphenol and flavonoid content is the highest in the extract of *C. orientalis*, and the lowest in that of *C. phrygia*. Similarly, *C. orientalis* shows the greatest scavenging activity on DPPH (1,1-diphenyl-2-picryl-hydrazyl), ABTS [2,2′-azobis(3-ethylbenzothiazoline-6-sulfonate)] and Fe^3+^ reducing power assays, whereas the lowest activity is found for *C. phrygia*. The highest content of reducing sugars is found in *C. nigra*, while *C. orientalis* has the highest levels of free amino acids. We find a strong positive correlation between total phenolics and flavonoids and the antioxidant capacity of all three *Centaurea* species. Moreover, the content of free amino acids strongly and positively correlates with the levels of total phenolics and flavonoids, antioxidant activity assessed by DPPH and ABTS assays and Fe^3+^ reducing power. Summing up, *C. orientalis* exhibits the strongest antioxidant potential of the investigated *Centaurea* species. This species could potentially be a natural source of antioxidant substances for the pharmacy, cosmetics and food industries. The content of free amino acids may be used as a marker of the antioxidant status of *Centaurea* species.

## 1. Introduction

Free radicals and antioxidants are natural components of the body, and are essential for the proper course of many metabolic processes. As long as they are in equilibrium, the body functions properly [1]. Oxidative stress occurs when the pro-oxidant (e.g., free radicals) and antioxidant factors do not balance each other [2]. The oxidative stress may damage cellular structures, accelerate the body’s aging and facilitate the development of many civilization diseases, such as obesity, diabetes, atherosclerosis, stroke, Parkinson’s disease, Alzheimer’s disease and cancer [3]. The ways of counteracting oxidative stress, e.g., various types of antioxidants, are intensively researched. Antioxidants help to protect the body against oxidative stress, as they neutralize free radicals and also control the functioning of antioxidant and detoxification enzymes [4]. Excessive amounts of synthetic antioxidants used as food preservatives, such as butyl derivatives of hydroxyanisole and hydroxytoluene and tertiary butylhydroquinone, do not reduce harmful effects of free radicals, but often contribute to their formation and greater activity, which may result in the initiation of diseases [5,6]. For these reasons, many studies focus on finding non-toxic, natural sources of antioxidants [7]. Products and raw materials of plant origin, abundant in antioxidants, have enormous antioxidant potential. The most important compounds include polyphenols that can inhibit or delay oxidation processes in both living organisms and food [8,9].

The genus *Centaurea* of Asteraceae family includes over 600 species, mainly from the Middle East and the Mediterranean region [10,11]. Some of them have long been used as medicinal plants in folk tradition. The herbal raw material of various *Centaurea* species has been used in the treatment of renal dysfunction, inflammation of the urinary tract and gynecological, digestive and dermatological conditions [12]. Studies show that extracts and essential oils obtained from some *Centaurea* species may exhibit anti-cancer, anti-diabetic, anti-inflammatory, analgesic, anti-rheumatic, hepatoprotective, antioxidant and antimicrobial properties [13,14]. In addition to their phytotherapeutic application, *Centaurea* species are used in cosmetics, in industry as a source of dyes, and in gastronomy as edible flowers [15]. The most important groups of active compounds found in *Centaurea* species involve polyphenols (including flavonoids and lignans) and sesquiterpene lactones. In most studies, herbal raw materials of *Centaurea* species comprised shoots with leaves collected during flowering [16,17,18]. The properties of extracts obtained from *Centaurea* flowers alone are less well known [19]. The raw material most widely used in herbal medicine, cosmetology and food industry includes edible flowers of *C. cyanus* and, more specifically, the upper parts of their capitulum [20]. However, other less known *Centaurea* species, cultivated as ornamental plants in wildlife gardens and wildflowers, also deserve attention. They include *C. nigra* L. with purple inflorescences and blackish involucres, *C. orientalis* L. with bright yellow inflorescences and *C. phrygia* L. with red–purple inflorescences. These species are perennials with low soil requirements, preferring dry and sunny locations [21]. *C. nigra* has been a popular medicinal plant since middle ages. The petals, roots and seeds of *C. nigra* have been used to relieve throat inflammation, stop gingival bleeding, accelerate wound healing and as a diuretic and tonic [22]. Kenny et al. [23] classified *C. nigra* as a wild edible plant and found that its extracts not only possess antioxidant properties, but also show effective antimicrobial activity against Gram-positive bacterial strains. Research literature lacks data on biological properties, including the antioxidant activity of *C. orientalis* and *C. phrygia*.

The aim of this work is to compare the total content of polyphenols and flavonoids as well as the antioxidant activity of flower extracts from *C. nigra*, *C. orientalis* and *C. phrygia*. We also determine the flower content of free amino acids and reducing sugars. We assume that the flowers of the investigated *Centaurea* species can be a potential source of antioxidants.

## 2. Results and Discussion

Polyphenols are secondary plant metabolites with antioxidant properties resulting from the presence of the –OH (hydroxyl) group, scavenge free radicals [24]. In addition to their antioxidant activity, plant polyphenolic compounds have antiproliferative, antibacterial, anti-inflammatory and anti-allergic properties [25]. The antioxidant activity of plant extracts usually correlates with the content of polyphenols, both in terms of individual active substances and their total content [26]. Therefore, the assessment of total phenolic and total flavonoid content in plant extracts is an important step in the evaluation of herbal raw materials. This applies in particular to complex mixtures, which often comprise compounds exerting synergistic effects and enhancing the action of biologically active components [27,28]. In our experiment, the approximate content of phenolics in the flower extracts of *C. nigra*, *C. orientalis* and *C. phrygia* was assessed with the Folin–Ciocalteu reagent and the results were expressed in mg of gallic acid equivalent per gram of extract (mg GAE/g of extract). Figure 1 shows significant differences in the total phenolic content in individual *Centaurea* species. The highest concentration of polyphenols was detected in *C. orientalis* (19.38 mg GAE/g), then in C. *nigra* (12.0 mg GAE/g) and, finally, *C. phrygia* (8.08 mg GAE/g). Kenny et al. [23] analyzed aqueous and ethanolic extracts from the aerial parts of *C. nigra* and reported the total polyphenol content to range from 61.7 mg GAE/g to 173.2 mg GAE/g. In a comparative study of *Centaurea aksoyi* and *Centaurea amaena* total phenolics, the extracts ranged from 3.28 to 12.92 mg GAE/g [29]. A considerably higher content of total phenolics, reaching from 82.27 to 175.40 mg GAE/g was reported for herbal extracts of *Centaurea kurdica*, *Centaurea rigida*, *Centaurea amanicola*, *Centaurea cheirolopha* or *Centaurea ptosimopappoides* [16].

The largest and best known group of phenolic compounds is flavonoids [30]. They are found mainly in flowers, fruits and leaves, and their names are derived from their colors, which can be yellow, blue, red and purple [31]. Figure 2 shows the total content of flavonoids in flower extracts of individual *Centaurea* species expressed as mg of quercetin equivalents (QE) per gram of extract. As in the case of polyphenols, the extracts richest in flavonoids (40.57 mg QE/g) were those derived from *C. orientalis*. Significantly lower flavonoid levels were found in *C. nigra* (28.57 mg QE/g) and *C. phrygia* (24.92 mg QE/g). These values were higher than those reflecting the total flavonoid content in *C. aksoyi* (5.83 mg QE/g) and *C. amaena* (7.69 mg QE/g) [29]. In another study [17] comparing the biological activity of *Centaurea* species, where the flavonoid content was expressed as rutin equivalents (RE), flavonoid concentration in aqueous and methanolic extracts amounted to, respectively, 30.76 and 46.76 mg RE/g in *Centaurea antalyense*, 40.38–45.94 mg RE/g in *Centaurea polpodiifolia* var. *pseudobehen* and 55.75 and 29.43 mg RE/g in *Centaurea pyrrhoblephara*. It should be remembered that comparing the total polyphenol and flavonoid content determined in various studies and converted with different formulas is subject to error. The results can be significantly affected by the sample preparation method, interferences or environmental factors that determine the composition and properties of plant raw materials [26,30]. Genotypes differ in the composition and content of phenolic compounds, which can be used to distinguish and identify them [32]. 

None of the commonly used biochemical methods offered an effective assessment of the antioxidant activity of all compounds present in a given plant extract [33,34]. Therefore, the antioxidant capacity of extracts should be tested using various methods, preferably based on different mechanisms of action. In our study, the antioxidant capacity of flower extracts from *C. nigra*, *C. orientalis* and *C. phrygia* was established in vitro based on free radical scavenging activity (DPPH and ABTS) and Fe^3+^ reducing power. In the DPPH method the antiradical activity was determined based on the ability of antioxidants to scavenge the model 1,1-diphenyl-2-picryl-hydrazyl (DPPH) radical. As the compound is soluble only in organic solvents, the method mainly measures the potential of lipophilic antioxidants [35]. The ABTS method is routinely used to determine the total antioxidant activity of both hydrophobic and hydrophilic antioxidant samples, and the radical is ABTS [2,2′-azobis(3-ethylbenzothiazoline-6-sulfonate)] [36]. The assessment of Fe^3+^ reducing power involves the determination of the antioxidant potential in a reaction mixture based on the reduction in Fe^3+^ to Fe^2+^ [37]. The antioxidant capacity of *Centaurea* species is summarized in Table 1. The greatest capacity to scavenge the DPPH radical was confirmed for the extracts of *C. orientalis* (39.34%). It was significantly lower in *C. nigra* (10.47%) and the lowest in *C. phrygia* (9.43%). *C. orientalis* extract also showed the highest ABTS radical scavenging efficiency (91.69%), while in the extracts of *C. nigra* and *C. phrygia*, the antioxidant potential reached 44.62% and 39.69%, respectively. The antioxidant activity of the extracts measured with Fe^3+^ reducing power brought about similar results. The extracts of *C. orientalis* exhibited a nearly two times greater reducing power (1.77) than those of *C. nigra* (0.84) and *C. phrygia* (0.74). Research shows [16,29,38] that the antioxidant potential of various *Centaurea* species depends on the analytical method used, the extraction time and the type of solvent and its concentration. For example, in *C. polypodiifolia* var. *pseudobehen*, *C*. *pyrrhoblephara* and *C. antalyense*, the values yielded by the ABTS assay ranged were from 16.32% to 93.42%, for DPPH from 12.97% to 93.53% and for the Fe^3+^ reducing power from 0.17 to 1.17 [17].

A comparison of the results yielded by these two methods revealed a few times higher values for the ABTS assay than for the DPPH test, which may have depended on the solubility of the radical in water and organic solvents [36]. Similarly, Aktumsek et al. [17] reported a higher activity of methanolic extracts from *C. polypodiifolia* var. *pseudobehen*, *C. pyrrhoblephara* and *C. antalyense* herb measured by the ABTS rather than the DPPH method. Additionally, an analysis of the antioxidant potential of *C. nigra* extracts showed stronger antioxidant activity when assessed by the FRAP rather than the DPPH test [23]. 

Interestingly, *C. orientalis* seemed much more efficient than the other two species in removing DPPH and ABTS radicals and showed a greater Fe^3+^ reducing power. This could be explained by a greater content of total polyphenols and flavonoids in *C. orientalis* tissues. Many studies confirmed that the ability to scavenge free radicals closely correlated with the total content of polyphenols and flavonoids [26,39,40]. Our findings corroborated this conclusion, as we found a very close correlation between the total polyphenols and antioxidant activity, i.e., *r* = 0.91 for DPPH, *r* = 0.93 for ABTS and *r* = 0.94 for the Fe^3+^ reducing power. Similarly, a very high correlation coefficient was found for the flavonoid content and the results of DPPH (*r* = 0.85) and ABTS (*r* = 0.89) tests and the Fe^3+^ reducing power (*r* = 0.90) (Table 2). Equally high correlation coefficients, ranging from *r* = 0.76 to *r* = 0.97 between the total phenolic content and total flavonoid content and total antioxidant capacity, inhibition capacity, Fe^3+^ reducing power, free radical scavenging activity and cupric reducing antioxidant capacity, were calculated for *Centaurea pseudoscabiosa* subsp. *araratica*, *Centaurea pulcherrima* var. *pulcherrima*, *Centaurea salicifolia* subsp. *abbreviata* and *Centaurea babylonica* [41].

For several years, edible flowers have been the subject of intensive research for their health-promoting and nutritional properties, as many of them are a source of various bioactive and nutritional compounds [42]. Our study revealed significant differences in the content of reducing sugars and free amino acids in the flowers of investigated *Centaurea* species (Table 3). The highest level of reducing sugars (189.52 mg/g) was detected in *C. nigra*, while the flowers of *C. orientalis* were the richest in free amino acids (4.69 mg Gly/g). Fernandes et al. [43] analyzed the nutritional value of *C. cyanus* flowers, and found that its petals contained 3.43 g/100 g FW fructose, 7.30 g/100 g FW glucose, 1.18 g/100 g FW sucrose and 1.66 g/100 g FW protein. The three investigated *Centaurea* species demonstrated a significant and very close correlation between the content of free amino acids, total polyphenols (*r* = 0.96) and total flavonoids (*r* = 0.91) (Table 4). Moreover, the content of free amino acids also correlated with the results of antioxidant activity determined by the DPPH (*r* = 0.82) and ABTS (*r* = 0.86) assays and the capacity to reduce Fe^3+^ (*r* = 0.87) (Table 2). Amino acids and peptides are compounds that can act as antioxidants by breaking down peroxides, quenching singlet oxygen or chelating metals [44,45]. The close correlations between the content of free amino acids and total phenolic and flavonoid concentrations, the capacity to remove DPPH and ABTS radicals and Fe^3+^ reducing power suggests that the level of free amino acids may be used as an indicator of antioxidant properties of *Centaurea* species.

## 3. Materials and Methods

### 3.1. Plant Materials

Seed-propagated three-year-old plants of *C. nigra*, *C. orientalis* and *C. phrygia* were cultivated on research plots belonging to the West Pomeranian University of Technology in Szczecin. They grew on a sunny site, in sandy soil containing N-NO_3_ 53 mg/dm^3^, P 44 mg/dm^3^ and K 99 mg/dm^3^. Soil pH determined in water was 7.1. The plants were fertilized twice in the spring with Azofoska, at a dose of 30 g of the fertilizer per m^2^. The flowers were collected in the morning of 7 July 2020 from plants in full bloom. The analyses involved only colored petals, without lower parts of the capitulum (Figure 3). The plant material was dried immediately after harvesting in a dark and airy place at 22–25 °C for seven days, and then stored in a tightly closed vessel in the dark at 18–20 °C.

### 3.2. Preparation of Extracts

Flower extracts were prepared as described by Grzeszczuk et al. [46] with modifications. The dried plant material was ground in a grinder, and then the homogenized material (0.5 g) was supplemented with 40 mL of 80% methanol and deionized water in 7:3 ratio (*v*/*v*). The samples were placed in an ultrasonic bath for 30 min. The extracts were centrifuged for 5 min at 5000× *g* (Centrifuge 5418 Eppendorf, Warsaw, Poland) and filtered through 0.22 µm nylon membrane filters (Merck, Darmstadt, Germany). All the extractions were prepared in triplicate. The final extracts were stored at −20 °C. 

### 3.3. Determination of Total Polyphenols and Total Flavonoids

Total polyphenol content of each extract sample was determined by Folin–Ciocalteu method as described by Tong et al. [47] with own modifications. To this end, 20 μL of the extracts was mixed with 150 μL of distilled water and 100 μL of Folin–Ciocalteu reagent. After 5 min, the mixture was supplemented with 80 μL of Na_2_CO_3_ (7.5%), and then incubated for 30 min at 40 °C in the dark. Absorbance was measured at 765 nm with a microplate reader (Synergy LX, BioTek, Winooski, VT, USA). Gallic acid (GAE) was used to prepare the standard curve and the results were expressed as milligrams GAE per g extract. The content of flavonoids was determined as described by Tong et al. [48] with small modifications. To this end, 250 μL of the extracts was mixed with 100 μL of distilled water and 7.5 μL of NaNO_2_. After 5 min, the mixture was supplemented with 7.5 μL of 10% AlCl_3_ solution. The samples were left for 6 min at room temperature before the addition of 25 µL of 1 M NaOH. Then, the mixture was diluted with 135 μL of distilled water and absorbance was measured at 510 nm. Quercetin was used for the preparation of the calibration curve and the results were expressed as mg quercetin equivalent (QE) per g extract.

### 3.4. Determination of Fe^3+^ Reducing Power

The reducing capacity of each extract sample was measured as described by Łopusiewicz et al. [48]. The extracts (500 µL) were placed in 1.5 mL Eppendorf tubes to which 1.25 mL of phosphate buffer (0.2 M, pH 6.6) and 1.25 mL of 1% potassium hexacyanoferrate were added. The samples were incubated at 50 °C for 20 min before addition of 1.25 mL of trichloroacetic acid. Centrifugation at 3000× *g* for 10 min (Centrifuge 5418 Eppendorf, Warsaw, Poland) yielded 1.25 mL of the supernatant. Then, 0.25 mL of a 0.1% iron chloride was added and, finally, absorbance was measured at 700 nm to determine the reducing power.

### 3.5. DPPH and ABTS Assays

DPPH and ABTS radical scavenging activity was measured according to the procedures described by Łopusiewicz et al. [48]. For DPPH assay, 0.5 mL of the extracts was mixed with 0.5 mL of 0.01 mM methanolic solution of DPPH. The samples were incubated in the dark for 30 min. The absorbance was measured at 517 nm. For ABTS assay, 3 mL of ABTS solution was mixed with 50 µL of the extracts and incubated in the dark for 6 min. Before the test, the ABTS solution was diluted in ethanol to the absorbance of 0.700 ± 0.02, and the absorbance was measured at 734 nm.

### 3.6. Determination of Reducing Sugars Content

The content of reducing sugars was determined by the DNS method (3,5-dinitrosalicylic acid) as described by Łopusiewicz et al. [48]. First, 0.5 mL of the extract was mixed with 0.5 mL of 0.05 M acetate buffer (pH 4.8), and 1.5 mL of DNS reagent, and the mixture was shaken vigorously. The solutions were incubated in boiling water for 5 min and then cooled to room temperature. The absorbance was measured at 540 nm. Glucose in acetate buffer was used to prepare the standard curve.

### 3.7. Determination of Free Amino Acids Content

The level of total free amino acids was determined as reported by Łopusiewicz et al. [49]. Briefly, 1 mL of the extracts and 2 mL of ninhydrin-Cd reagent were placed in test tubes. The solutions were vortexed, heated at 84 °C for 5 min and then cooled on ice. The absorbance was measured at 507 nm. Glycine (Gly) was used to prepare the standard curve.

### 3.8. Statistical Analysis

The analyzes were performed in triplicate. To assess the significance of the differences between the mean values, the one-way analysis of variance and Tukey’s test at the significance level of 0.05 were conducted. In addition, Pearson’s linear correlation coefficients were calculated for the selected parameters. Statistical calculations were determined using Statistica 13.3 (TIBCO Inc. Statsoft, Kraków, Poland) software.

## 4. Conclusions

Considering the potential of *Centaurea* genus plants in the medicine, pharmacy, cosmetics and food industries, the research on the antioxidant properties of their less known taxa seems fully justified. This experiment showed a variable antioxidant potential of flower extracts derived from *C. nigra*, *C. orientalis* and *C. phrygia. C. orientalis* was found to be the richest in total polyphenols, total flavonoids and free amino acids, and it also exhibited the strongest antioxidant properties. This was the first study indicating a correlation between the content of free amino acids, polyphenols and the antioxidant capacity of *Centaurea* species. Further studies are necessary to identify specific compounds (phenolics, flavonoids and amino acid profiles) responsible for the antioxidant activity of the investigated *Centaurea* species.

## Figures and Tables

**Figure 1 molecules-26-07465-f001:**
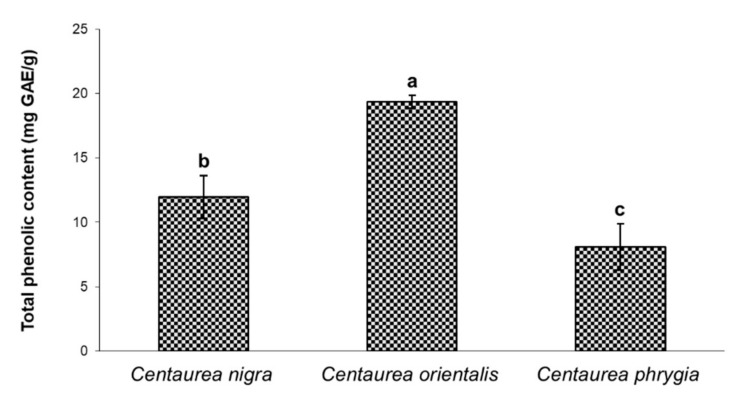
Total phenolic content (mg gallic acid equivalent (GAE) per gram of extract) of three *Centaurea* species. The data represent the mean ± standard deviation of three replicates. Different letters above bars indicate significant differences at *p* < 0.05 (Tukey LSD test).

**Figure 2 molecules-26-07465-f002:**
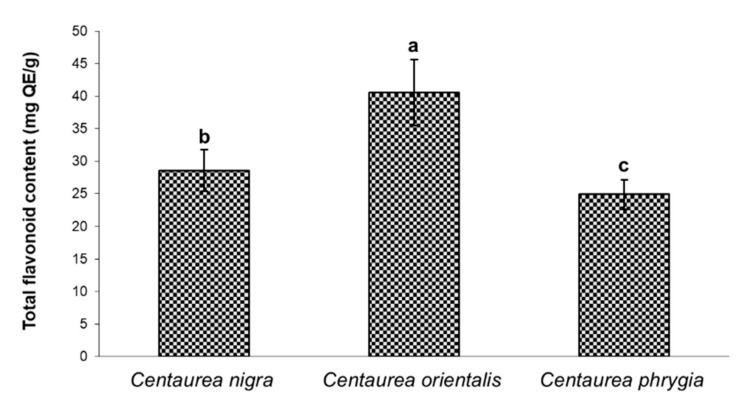
Total flavonoid content (mg quercetin equivalent (QE) per gram of extract) of three *Centaurea* species. The data represent the mean ± standard deviation of three replicates. Different letters above bars indicate significant differences at *p* < 0.05 (Tukey LSD test).

**Figure 3 molecules-26-07465-f003:**
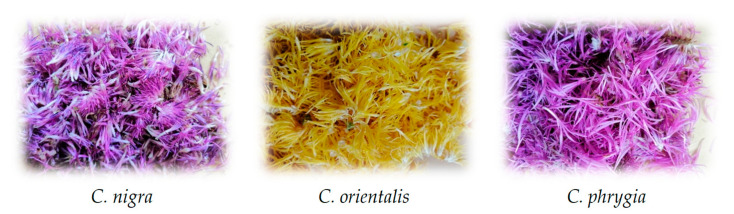
Visual appearance of three *Centaurea* species petals.

**Table 1 molecules-26-07465-t001:** DPPH radical scavenging activity (%), ABTS radical scavenging activity (%) and Fe^3+^ reducing power of three *Centaurea* species. The data represent the mean ± standard deviation of three replicates.

Species	DPPH (%)	ABTS (%)	Fe^3+^ Reducing Power
*C. nigra*	10.47 ± 0.84 ^b1^	44.62 ± 0.44 ^b^	0.84 ± 0.01 ^b^
*C. orientalis*	39.34 ± 2.24 ^a^	91.69 ± 0.85 ^a^	1.77 ± 0.01 ^a^
*C. phrygia*	9.43 ± 0.89 ^b^	39.69 ± 1.95 ^c^	0.74 ± 0.02 ^c^

^1^ Means over each column not marked with the same letter are significantly different at *p* < 0.05 (Tukey LSD test).

**Table 2 molecules-26-07465-t002:** Pearson correlation analysis between DPPH, ABTS and Fe^3+^ reducing power and total phenolic content, total flavonoid content, reducing sugars and total free amino acids of three *Centaurea* species.

Compound	DPPH	ABTS	Fe^3+^ Reducing Power
Total phenolic content	0.91 *	0.93 *	0.94 *
Total flavonoid content	0.85 *	0.89 *	0.90 *
Reducing sugars	−0.51	−0.47	−0.47
Total free amino acids	0.82 *	0.86 *	0.87 *

* Correlation is significant at the 0.01 level (2-tailed) based on Pearson’s correlation coefficient method.

**Table 3 molecules-26-07465-t003:** Reducing sugars and total free amino acids of three *Centaurea* species. The data represent the mean ± standard deviation of three replicates.

Species	Reducing Sugars (mg/g)	Total Free Amino Acids (mg Gly/g)
*C. nigra*	189.52 ± 3.98 ^a1^	3.72 ± 0.25 ^b^
*C. orientalis*	151.11 ± 1.71 ^b^	4.69 ± 0.26 ^a^
*C. phrygia*	153.26 ± 1.51 ^b^	2.79 ± 0.25 ^c^

^1^ Means over each column not marked with the same letter are significantly different at *p* < 0.05 (Tukey LSD test).

**Table 4 molecules-26-07465-t004:** Pearson correlation analysis of total phenolic content, total flavonoid content, reducing sugars and total free amino acids of three *Centaurea* species.

Compound	Total Phenolic Content	Total Flavonoid Content	Reducing Sugars	Total Free Amino Acids
**Total phenolic content**	1			
**Total flavonoid content**	0.91 *	1		
**Reducing sugars**	−0.20	−0.29	1	
**Total free amino acids**	0.96 *	0.91 *	−0.03	1

* Correlation is significant at the 0.01 level (2-tailed) based on Pearson’s correlation coefficient method.

## Data Availability

The data presented in this study are available within the article.

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
