# Peer review of "Antioxidant Potential of Flower Extracts from Centaurea spp. Depends on Their Content of Phenolics, Flavonoids and Free Amino Acids"

_molecules, 2021, doi:10.3390/molecules26247465_

Round 1

Reviewer 1 Report

The paper was well-organized for methodology and analytical approach. Moreover  it appear interesting for the new knowledge concerning the flower extracts of C. nigra L., C. orientalis L. and C. phrygia L.

The English form is fit to the publication and the paper is useful for publication after a minor revision, as follows:

Please discuss the reason why flavonoids are not congruent with the quantity of observed polyphenols in our study. Report possible justifications to it.

Author Response

Thank you very much for your time spent on a careful and detailed revision of our manuscript. We are grateful for numerous comments and remarks that made us reconsider many fundamental issues. Invaluable content and style related corrections allowed us to avoid multiple mistakes. In general, the review let us considerably improve our manuscript. Below you will find our answers to all the remarks. We hope our explanations are comprehensive and will dispel any possible doubts.

The paper was well-organized for methodology and analytical approach. Moreover  it appear interesting for the new knowledge concerning the flower extracts of C. nigra L., C. orientalis L. and C. phrygia L.

Response: We are thankful to the reviewer for the positive comments.

The English form is fit to the publication and the paper is useful for publication after a minor revision, as follows: Please discuss the reason why flavonoids are not congruent with the quantity of observed polyphenols in our study. Report possible justifications to it.

Response: We are thankful to the reviewer for the positive comments. The reason for the differences between polyphenol and flavonoid content is the two different methods of determination. For polyphenols, the Folin-Ciocalteu and standard gallic acid methods were used. For flavonoids, the aluminum chloride method was used. As mentioned in our work (Lines 128-132) the differences in methods may be due to the use of different standards e.g. rutin or quercetin. Different contents of polyphenols and flavonoids were also reported in other reports which have been quoted in our work (References 17,23,29). 

Reviewer 2 Report

The article needs to be thoroughly revised for the English language.

There are many typos/syntax errors:

Title: give space between aminoacids (line 4)

Word eliminate is used at many places instead of scavenge.

Conclusion is missing in the abstract.

Sec 3.2: you mentioned leaf extract but the study is on flowers. (line 225)

Which phenolic compounds are present in these species. Quantify them by HPLC.

Before using an acronym/abbreviation, expand it in the first place of occurrence. in the text. 

Why TPC and TFC were determined in only one extract??

No positive standard is used to compare the antioxidant activity. 

Author Response

Reviewer 2

Thank you very much for your time spent on a careful and detailed revision of our manuscript. We are grateful for numerous comments and remarks that made us reconsider many fundamental issues. Invaluable content and style related corrections allowed us to avoid multiple mistakes. In general, the review let us considerably improve our manuscript. Below you will find our answers to all the remarks. We hope our explanations are comprehensive and will dispel any possible doubts.

The article needs to be thoroughly revised for the English language. There are many typos/syntax errors: Title: give space between aminoacids (line 4). Word eliminate is used at many places instead of scavenge.

Response: Thank you for your comment. We are grateful to the reviewer for pointing out our mistakes, which are very helpful for us to improve the manuscript. As suggested, the whole manuscript has been re-proofread to ensure the best possible version.

Conclusion is missing in the abstract.

Response: Thank you for your comment. Abstract has been modified and conclusion have been highlighted.

Sec 3.2: you mentioned leaf extract but the study is on flowers. (line 225)

Response: Thank you for your comment. We are sorry for this mistake. We have modified “leaf” in “flower”.

Which phenolic compounds are present in these species. Quantify them by HPLC.

Response: Thank you for your comment. This suggestion is very valuable, so we will continue to explore these additional components in subsequent study. We inform about it in the conclusion.

Before using an acronym/abbreviation, expand it in the first place of occurrence. in the text. 

Response: Thank you for your comment. We gave full abbreviations.

Why TPC and TFC were determined in only one extract??

Response: Thank you for your comment. All the extractions were prepared in triplicate.

No positive standard is used to compare the antioxidant activity. 

Response: Thank you for this comment. In fact, we used the relative radicals scavenging capacity comparison approach to compare the potential of Centaurea species among themselves. In our further in depth studies, certainly these analyses will be deepened by reference to known antioxidants (such as trolox and ascrobic acid) to be able to express the antioxidant potential in their equivalents.

Reviewer 3 Report

Journal: Molecules

Manuscript ID: molecules-1493550

Type of manuscript: Communication

Title: Antioxidant Potential of Flower Extracts from Centaurea spp. Depends on their Content of Phenolics, Flavonoids and Free Aminoacids

Authors: Piotr Salachna *, Anna Pietrak, Łukasz Łopusiewicz Submitted to section: Natural Products Chemistry

The manuscript focused on the antioxidant potential Centaurea spp. and further discussed the relation between antioxidant potential phytoconstituents. The manuscript describes the Centaurea spp. have great antioxidant potential. The authors have presented the article in a very nice way; the work done is to the mark of the journal. I suggest the following suggestions for the betterment of the article.

  1. In the figure legends and Table titles, the sentence -The date represent the mean…… must be the data represents the mean………
  2. The authors must provide the correlation data based on which the conclusion has been made. Therefore, add the correlation between antioxidant potential and Phenolics, Flavonoids, and Free Amino acids content.
  3. The Pearson linear correlation was done (Line 280-81) but authors must include the figure or Table of correlation and then discuss the results.

The work done is up to the mark, therefore I suggest a minor revision of the article.

Author Response

Thank you very much for your time spent on a careful and detailed revision of our manuscript. We are grateful for numerous comments and remarks that made us reconsider many fundamental issues. Invaluable content and style related corrections allowed us to avoid multiple mistakes. In general, the review let us considerably improve our manuscript. Below you will find our answers to all the remarks. We hope our explanations are comprehensive and will dispel any possible doubts.

The manuscript focused on the antioxidant potential Centaurea spp. and further discussed the relation between antioxidant potential phytoconstituents. The manuscript describes the Centaurea spp. have great antioxidant potential. The authors have presented the article in a very nice way; the work done is to the mark of the journal.

Response: Thanks for your kind comment.

I suggest the following suggestions for the betterment of the article. In the figure legends and Table titles, the sentence -The date represent the mean…… must be the data represents the mean………

Response: Thank you for your comment. We are sorry for this mistake. We have modified “date” in “data”.

The authors must provide the correlation data based on which the conclusion has been made. Therefore, add the correlation between antioxidant potential and Phenolics, Flavonoids, and Free Amino acids content. The Pearson linear correlation was done (Line 280-81) but authors must include the figure or Table of correlation and then discuss the results.

Response: Thank you for your advice. We have revised the correlation date according to your comment. The correlation coefficients are shown in the additional two tables.

The work done is up to the mark, therefore I suggest a minor revision of the article.

Response: Once again, thanks for your kind comment.

Round 2

Reviewer 2 Report

The authors have incorporated some of the suggested changes in the MS.